# Gastric Inflammation Impacts Serotonin Secretion in a Mouse Model of *Helicobacter pylori* Vaccination [note 1]

**DOI:** 10.3390/ijms26167735

**Published:** 2025-08-10

**Authors:** Sulaimon Idowu, Kate Polglaze, Thi Thu Hao Van, Robert J. Moore, Paul A. Ramsland, Paul P. Bertrand, Anna K. Walduck

**Affiliations:** 1School of Science, RMIT University, Bundoora, VIC 3083, Australia; sidowu@csu.edu.au (S.I.); kate.polglaze@gmail.com (K.P.); thithuhao.van@rmit.edu.au (T.T.H.V.); rob.moore@rmit.edu.au (R.J.M.); paul.ramsland@rmit.edu.au (P.A.R.); paul.bertrand@rmit.edu.au (P.P.B.); 2Rural Health Research Institute, Charles Sturt University, Orange, NSW 2800, Australia; 3Department of Immunology, Monash University, Melbourne, VIC 3004, Australia; 4Department of Surgery, Austin Health, The University of Melbourne, Heidelberg, VIC 3084, Australia

**Keywords:** *H. pylori*, vaccine, serotonin, microbiota

## Abstract

*Helicobacter pylori* infection causes inflammation in the gastric mucosa, and this has been reported to disrupt the gastric microbiota. Serotonin (5-HT) is a key neurotransmitter in the gut–brain axis and plays key roles in intestinal homeostasis and immune function. We investigated gastric serotonin release in *H. pylori*-infected mice and observed increased release in vaccinated, challenged mice compared to sham vaccinated controls. We investigated the effects of 5-HT on epithelial responses in an in vitro human gastric cancer cell line model (AGS), as well as inflammatory responses and the gastric microbiota in a C57BL/6 mouse model of *H. pylori* infection. HTR1A was upregulated in the stomachs of mice chronically infected with *H. pylori* SS1 (3 weeks) compared to uninfected controls, whereas HTR2B was upregulated only in acutely infected mice (3 days), consistent with a role for 5-HT signalling in the development of gastritis. Exposure to 5-HT did not affect NF-κB activation in *H. pylori*-exposed AGS cells but did inhibit extracellular signal-regulated kinase 1 (ERK1) translocation. Analysis of the gastric microbiota revealed that while vaccination did not significantly affect the diversity of the microbiota, vaccinated animals had increased abundance of *Lactobacilli*. Our results suggest that local inflammation caused by *H. pylori* is responsible for increased 5-HT release.

## 1. Introduction

The immune response to *H. pylori* infection is characterised by an unresolved inflammation of the gastric epithelium which later manifests into diseases such as peptic ulcers or gastric cancer [1]. This chronic inflammation involves tightly regulated innate and adaptive immune responses; however, given the prevalence of infection, it is clear that the natural immune response is not effective for clearance in the majority of cases [2]. Vaccination against the pathogen in mice results in 1–2 log reductions in colonisation, but it also increases local inflammation, at least in the initial weeks following vaccination and is termed post-immunisation gastritis [3]; however, the protective mechanisms are not well understood [2,4,5,6].

Gut hormones are increasingly being recognised for functioning both as hormones and exhibiting cytokine-like behaviour in the gut [7,8]. Several studies have reported the anti-inflammatory and pro-inflammatory effects of the gastric hormones ghrelin and leptin [9,10,11,12,13,14]. Serotonin, also known as 5-hydroxytryptamine (5-HT), has also been reported to mediate the release of cytokines and chemokines in the gut [7,15]. The participation of intestinal hormones in the local immune response has led to the study of their application for therapeutic purposes in chronic inflammatory disease of the gut, such as in mouse models of colitis and acute pancreatitis [16,17,18,19]. Thus, modulating the actions of gut hormones in inflammation has potential as a therapeutic approach.

The innate immune response to *H. pylori* infection is characterised by inflammation mainly via the activation of the NF-κB pathway [20,21,22], leading to the upregulation of interleukin-8 (IL-8) in humans and the murine homologues CXCL1 (KC/MIP-1) and CXCL2 (MIP-2) in the gastric mucosa [23,24]. Although there is evidence to suggest that increases in serotonin production may be part of the host cell response to the activation of the NF-κB pathway stimulated by *H. pylori* infection [25,26], there have been no reports to date to suggest a role of serotonin on the local immune response to *H. pylori*.

The biological functions of serotonin are mediated via binding to cell surface receptors known as 5-HT receptors [27,28]. These receptors are widely expressed on immune cells, Paneth cells, goblet cells, and enterocytes, with each cell mostly expressing more than one receptor subtype (reviewed in [29]). Serotonin signalling through these receptors can have a varying effect on immune cell populations. For example, the 5-HT1A and 5-HT7 receptors have been reported to mediate inflammation via NF-κB pathways in macrophages and T cells [18,30,31,32]. However, the effect of serotonin on these pathways and the participation of these receptors in the gastric epithelium during *H. pylori* infection has not been elucidated.

The roles of gastric hormones in *H. pylori*-induced inflammation and their potential role in vaccine-induced protection have also been reported. In *H. pylori* infection, ghrelin exerts anti-inflammatory effects via the inhibition of the transcription factors of NF-κB and MAP kinases [33,34]. The requirement of leptin signalling for successful vaccine-induced protection against *H. pylori* has been reported in mice, as leptin receptor-deficient mice were not able to reduce colonisation post-vaccination [35].

An in-depth understanding of the factors responsible for *H. pylori*-induced inflammation will be needed to identify new targets for treatment and aid in the development of vaccines against the pathogen. An important factor in homeostasis and inflammation in the stomach is the role of the microbiota. The stomach has long been considered to be a “sterile organ” due to its low pH, and research on the gastric microbiota was only undertaken after initial reports of *H. pylori* as a cause for gastric disease. The advent of molecular methods permitted a better understanding of the unculturable microbiota, and an early study reported that the majority of sequences from gastric biopsy samples belonged to the *Proteobacteria*, *Firmicutes*, *Actinobacteria*, *Bacteroidetes*, and *Fusobacteria* phyla [36]. In the murine model, early studies on the effect of *H. pylori* infection on the gastric microbiota reported in one case that the microbiota was not significantly affected, while another study reported that diversity was reduced [37,38]. Differences in the detection methods and relatively low sensitivity of the methods may explain the differences between these studies. Interestingly, a study by Aebischer et al. reported that while the proportion of *Lactobacillus* spp. was reduced in *H. pylori*-infected mice compared to naïve mice, it was increased in vaccinated mice, suggesting that vaccination may restore a more favourable gastric microbiota [38]. In light of more recent studies that have established a link between gut microbiota and hormone release, including serotonin, the role of the microbiota in gastric inflammation should be reconsidered, including whether these effects may be site-specific, as microbiota-induced 5-HT biosynthesis was found to occur in colonic but not small intestinal enterochromaffin cells [39].

Prophylactic vaccination against *H. pylori* is successful in mouse models, resulting in reduced colonisation [40]. To date, vaccines trialled in humans have had limited success, and aside from promising outcomes in a field trial in China [41], there is currently no registered vaccine [40]. The mouse model is accepted as a good model of the inflammatory response to *H. pylori* and the roles of immune infiltrates in successful vaccination. Vaccination strategies against *H. pylori* include both oral and parenteral routes. A key characteristic of oral vaccines is the requirement of adjuvants or carriers. The safety and efficacy of the *Salmonella enterica* serovar Typhimurium strains expressing *H. pylori* urease A and B subunit have been demonstrated in mice [4,42], and an equivalent recombinant attenuated Ty21a Salmonella vaccine expressing *H. pylori* antigens was safe in human volunteers [43]. Furthermore, human recipients of the vaccine in a human challenge study developed CD4^+^ T cell-biassed responses, as also observed in mice [44].

This study investigated the role of serotonin on the local immune response to *H. pylori* infection, its effect on inflammation, and the participating serotonin receptors using in vitro and in vivo infection models. We proposed the hypothesis that 5-HT acts directly on gastric epithelial cells through interaction with serotonin receptors present on the epithelium, exerting pro-inflammatory effects in response to *H. pylori* infection via the NF-κB pathway. The expression of 5-HT receptors in the gastric mucosa of vaccinated and control mice was investigated, as well as a potential link to the gastric microbiota.

## 2. Results

### 2.1. Reduced H. pylori Colonisation and Increased Inflammatory Infiltrates in Gastric Mucosa of Vaccinated Mice

*H. pylori* colonisation in gastric tissue at day 21 post-challenge was determined from half the stomach by qPCR. In keeping with previous studies using this model, vaccinated mice had significantly reduced *H. pylori* colonisation (Mann–Whitney U-test, *p* < 0.05, Figure 1A). Mice that received the S. Typhimurium carrier strain had no protection compared to the control. Half the stomach from three mice per group were embedded for cryosectioning and H&E staining. Gastritis was scored as previously described [45] (Figure 1B,C). Gastritis was characterised by the infiltration of lymphocytes and polymorphonuclear cells. The inflammatory cell infiltrates were visibly denser in vaccinated mice (Figure 1B, panels a–c), while the naïve mice (Figure 1B, Panel d) showed intact mucosa. Gastritis scores were increased in vaccinated mice in the corpus but not the antrum (Mann–Whitney U-test, *p* < 0.05, Figure 1C). A more intense infiltrate is correlated with reductions in bacterial colonisation.

### 2.2. Increased 5-HT Release in Corpus Region of Vaccinated Mice

Figure 2A shows representative gastric, duodenal, and colonic tissues mounted for electrochemical measurements. 5-HT release was measured at four sites in the stomach (Figure 2B), and in the duodenum, ileum, and the colon. 5-HT release was detected in all stomach regions tested, except the fundus, which had levels on average > 1 μM (Figure 2B,D). *H. pylori*-infected stomachs did not exhibit significantly increased levels of 5-HT compared to non-infected controls in any region of the stomach. 5-HT release was however significantly increased only in the lower corpus of vaccinated and carrier vaccinated mice compared to controls (Figure 2D), suggesting that the vaccine-induced 5-HT release is localised in this region of the stomach, noting that this region also exhibited the highest inflammation scores (Figure 1C). There was no increased 5-HT release in the carrier control group compared to naïve mice, further implying that the 5-HT release is antigen-specific and not attributable to the carrier strain alone. 5-HT release in the ileum and distal colon was also not affected by *H. pylori* infection or vaccination, indicating that the effects were local to gastric tissue. The increased release in the lower corpus of vaccinated mice is further supported by the visibly enhanced 5-HT peaks in electrochemical traces (Figure 2B), aligning with quantitative measurements.

### 2.3. H. pylori Infection Impacts Expression of 5-HT Receptors on Immune Cell Populations in the Gastric Mucosa

Quantitative PCR analysis of the expression of the 5-HT receptors HTR1A, HTR1B, and HTR2B in the stomachs of unvaccinated mice that had been acutely or chronically infected revealed that HTR1A expression was significantly increased in chronically infected mice and that HTR2B was increased in acutely infected mice (*p* < 0.05, Welches T-test) (Figure 3A). A more detailed analysis of inflammatory infiltrates in the stomachs of chronically infected vaccinated and control mice was undertaken using flow cytometry, where cells were successively gated based on forward and side scatter, size, surface expression of CD45 and CD3, and surface expression of CD4, CD11b, Ly6G, and HTR1A (Figure 3B). This gating strategy effectively distinguishes lymphoid and myeloid populations, allowing for a robust identification of HTRIA expression across distinct immune subsets. Amongst the immune cell populations, the highest proportion of HTR1A expression was exhibited by CD4^+^ T cells, suggesting a potentially more prominent serotonergic signalling role in T cell-mediated immune modulation compared to myeloid cells in the context of vaccination. Proportions of CD4^+^ T cells were found to be significantly higher in vaccinated mice and not carrier control mice compared to naïve mice (Figure 3C). This is consistent with previous reports [4] and also implies that the response is antigen-specific and not vector-related. An analysis of the infiltrates also revealed that CD4^+^ T cells, macrophages (CD3^−^, CD11b^+^), and neutrophils (CD3^−^, CD11b^+^, Ly6G^+^) had surface expression of HTR1A. Available antibodies directed against HTR1B and HTR2B were not successful in FACS. While vaccinated mice had increased infiltrates of CD4^+^ T cells (Figure 3C), a significantly lower proportion expressed HTR1A (Figure 3D), between 20 and 100% of neutrophils, and between 20 and 60% of macrophages expressed HTR1A, but there was not a significance difference between vaccinated mice and controls. Neutrophil counts exhibited a higher inter-individual variability, suggesting that regulation may be context-dependent or subject to local microenvironmental cues.

### 2.4. Effect of H. pylori Infection on Gastric Epithelial Cell Responses to 5-HT

The gastric epithelial cell line (AGS) was co-cultured with *H. pylori* 26695 for 6 h, and the expression of 5-HT receptors and IL-8 was determined by qPCR (Figure 4A). The expression of HTR1A, HTR1B, and HTR2B was detected, but exposure to bacteria did not significantly increase expression. Pretreatment with 5-HT for 5 min did not affect the induction of IL-8 mRNA expression by *H. pylori* after 6 h of infection (Figure 4B). HTR1A protein was expressed on AGS cells exposed to *H. pylori* for 6 h (Figure 4C), localising to both cytoplasmic and nuclear regions. As shown by nuclear translocation (Figure 4D), the activation of NF-κB occurred after exposure to *H. pylori* for 0.5 h, with or without 5-HT. Exposure to *H. pylori* induced NF-κB translocation, as did the PMA control, but not after treatment with 5-HT alone. 5-HT treatment did not increase *H. pylori*-induced NF-κB activation. As shown by nuclear translocation (Figure 4E), the activation of ERK occurred in AGS cells exposed to *H. pylori* for 4 h. Both PMA and *H. pylori* exposure caused nuclear translocation (white arrows), but translocation was blocked by combined 5-HT and *H. pylori* (x).

### 2.5. Vaccination Does Not Significantly Affect Composition of Gastric Microbiota

The microbiomes of the stomach of vaccinated and control mice were examined using 16S rRNA sequencing. Analysis of the relative abundance at the genus level showed that, although there was notable variation between individual animals at the day 21 timepoint, there was a bias toward *Lactobacilli* in vaccinated and *Allobaculum* in sham vaccinated mice (Figure 5A). There were no significant changes in alpha diversity at the genus level using the Chao1, Welch T-test/ANOVA method (Figure 5B), or Shannon diversity index, which considers both diversity and richness (Appendix A). Beta diversity was also not significantly affected (Appendix A). Inspection of the read data revealed that *Helicobacter* spp. were detected at low counts, mainly in the sham vaccinated group, but these reads were at low abundance and were filtered out of further analysis (Appendix A). Given that *H. pylori* were detected in the qPCR assay from the same samples (Figure 1A), it can be concluded that *H. pylori* was present at very low abundance compared to other microbial communities present in the samples.

Single factor analysis revealed that the *Clostridium sensu stricto* 1 had increased abundance in sham vaccinated mice (*p =* 0.00175) (Figure 5C). These are strict anaerobes and include pathogens and have been reported to be increased in the gastric microbiota of GC patients [36]. Core microbiome analysis at the genus level revealed a bias toward *Lactobacilli* (Figure 5D).

## 3. Discussion

The effect of *H. pylori* infection on epithelial pro-inflammatory signalling, gastric inflammation, and cancer has been well documented (reviewed in [46]). This chronic inflammatory response is, however, ineffective at clearing infection, leading to persistent infection and the long-term disease consequences of gastritis, ulcers, or gastric cancer. Prophylactic vaccination is effective in animal models and results in reduced colonisation, a result which to date has not translated to effective vaccines for humans [40]. Vaccination induces a qualitatively different immune response compared to persistent infection, and a better understanding of this response will assist in the development of more effective vaccines. This study has investigated the previously unaddressed question of the role of serotonin in *H. pylori* infection. Vaccination was found to lead to increased local serotonin release in gastric tissue, consistent with the known impact of 5-HT in inflammation and providing potential links to the gastric microbiota.

Given the known role of serotonin in intestinal function and its high level in the gut, we reasoned that the chronic inflammation in the stomach caused by *H. pylori* infection may impact serotonin release in distal regions of the intestine. Furthermore, given the previously reported roles for serotonin in chronic inflammatory disease such as Crohn’s disease [7,8,18,26,47] and the involvement of serotonin signalling in the pathogenesis of other enteric infections [48], 5-HT signalling might be expected to play a role in *H. pylori* gastritis.

The observation that *H. pylori* infection alone did not lead to significantly increased levels of 5-HT release was unexpected, but the increased levels observed in vaccinated and carrier vaccinated *H. pylori*-challenged stomachs was consistent with the observed increase in local immune infiltrates in the lower corpus of these mice (Figure 2D). This suggests that the increased release of 5-HT is related to local inflammation rather than an antigen-specific protective response, as carrier vaccinated mice did not have reduced colonisation. We detected an increased expression of HTR2B in acutely infected stomachs and HTR1A in chronically infected stomachs, suggesting that these may have mediated the effects, but as we were unable to localise the expression of receptors in immunohistochemistry using the available antibodies. We examined inflammatory infiltrates using flow cytometry and an in vitro epithelial cell model of *H. pylori* infection. We were able to detect the surface expression of HTR1A on CD4^+^ T cell, macrophage, and neutrophil populations. The expression of HTR1A was reduced on CD4^+^ T cells from vaccinated stomachs. CD4^+^ T cells are known to mediate vaccine-induced protection [49,50], and it has previously been reported that populations of T_reg_ cells that are usually prevalent in *H. pylori* gastritis are reduced in vaccinated mice [4]. These observations are consistent with reports that T_reg_ cells secrete significant amounts of 5-HT [51], which may represent a mechanism for the control of the inflammatory macrophage and neutrophil populations that expressed the HTR1A receptor. Further in vitro functional studies with isolated leukocytes would be required to confirm this mechanism.

Our investigation of the epithelial effects of 5-HT revealed that gastric epithelial cells expressed HTR1A and HTR1B and that mRNA expression was increased after exposure to *H. pylori*. Treating AGS cells with 5-HT did not enhance NF-κB activation (nuclear translocation of p65) induced by *H. pylori*, but it did block ERK translocation. The CagA toxin of *H. pylori* has been shown to bind to SHP2, activating pro-oncogenic phosphatase activity and Ras-ERK signalling [52]. The actions of 5-HT signalling may therefore act to abrogate this signalling, contributing to maintaining gastric homeostasis and preventing cancer development in most *H. pylori*-infected patients.

The impact of *H. pylori* infection on the gastric microbiota has previously been reported in adults, paediatric patients [53,54,55], and in mouse models [37,38]. This includes one report that vaccination led to normalisation of the reduced diversity induced by *H. pylori* infection, including the restoration of the dominance of *Lactobacillus spp* [38]. Increased presence of *Lactobaccillus* spp. in the stomach has been associated with increased *H. pylori* eradication [56,57]. Thus, the increased level of *Lactobacillus* spp. suggests additional protective functions of vaccination (Figure 6). In the current study, we analysed the effect of vaccination on the gastric microbiota with a consideration of more recent knowledge about the production of neurotransmitters by the microbiota [39,58,59]. Our analysis illustrates that, at least at the timepoint studied, while there was no significant change in alpha or beta diversity of the gastric microbiota in vaccinated mice compared to sham vaccinated mice (Figure 5B and Appendix A), the sham vaccinated mice displayed greater inter-individual variability, suggesting a stabilising effect of vaccination, where *Lactobacilli* remained dominant. Of interest, the sham vaccinated group had increased abundance of *Clostridium* spp., including known pathogens such as the *Clostridium sensu stricto* 1, which has been associated with increased inflammation and gastric cancer [60], further supporting the beneficial effects of vaccination.

We detected increased serotonin release in the stomach but not the colon of vaccinated mice (Figure 2D). A limitation of our analysis is that it does not permit us to draw conclusions on a role for the microbiota in stimulating local 5-HT release, but our results are in line with a previous report stating that colonic bacteria regulate local serotonin release [39]. The increased abundance of *Lactobacilli* in the gastric microbiota of vaccinated mice is consistent with a previous report [38] despite the lack of a significant effect on the overall diversity in the current study. Given the evidence that serotonin has effects on bacterial physiology (reviewed in [61]) and has even impacted the adhesion of the enteric pathogen *Campylobacter jejuni* in an in vitro study [62], it is possible that vaccine-induced serotonin release directly affected *H. pylori* in the stomach, although the concentrations of 5-HT used in our studies did not have an antimicrobial effect. Further studies at later timepoints after vaccination and comparisons of the effect on the gastric, intestinal, and colonic microbiota are required to determine whether the effects of vaccination result in further restoration of the microbiota.

In conclusion, our observations of increased 5-HT secretion provide potential explanations for previously observed phenomena in vaccinated mice—the reported role for 5-HT in barrier function and intestinal homeostasis may explain the increased gastric epithelial cell proliferation [63] reported in vaccinated mice. Furthermore, the known physiological effects of increased serotonin [47] in the gut would explain the minor adverse effects of nausea and diarrhoea reported from previous vaccination studies [44,64]. The observed increased 5-HT release in the gastric corpus of vaccinated mice is therefore likely to be a result of increased local inflammatory infiltrates rather than the actions of the gastric microbiota.

## 4. Materials and Methods

### 4.1. Mouse Model of H. pylori Infection

Eight-week-old female *Helicobacter*-free C57BL/6 mice were purchased from the Animal Resource Centre, Canningvale, Western Australia. Mice were fed a sterile standard diet, had access to water ad libitum, and were housed under specific pathogen-free conditions. All animal experiments were undertaken with approval from the RMIT Animal Ethics Committee under the Animal Ethics Committee approval number AEC 2021-24092-13648. All experiments and procedures were undertaken following approved animal handling and experimental protocols.

Mice (5–10 per group, sufficient to detect a difference of 0.5 log CFU at power of 0.95) were vaccinated on day 0 with a live recombinant-attenuated *Salmonella enterica* var Typhimurium (strain SL3261pYZ97 expressing UreA and B) by oral gavage, as previously described [4]. Control mice received the carrier *Salmonella* strain (SL3261) or PBS. Thirty-five days after vaccination, mice were challenged with a single oral dose of 3 × 10^8^  *H. pylori* SS1 strain using previously described methods [4]. Mice were euthanised on day 21 post-challenge, stomach tissue was removed, and gastric contents were carefully taken out and stored at −80 °C prior to DNA extraction for microbiota analysis. Stomach tissue was gently washed in cold phosphate buffered saline (PBS), rinsed with Ringer’s solution, and used for electrochemical measurements, as described below. From five mice per group, half of the tissue was homogenised to determine the CFU or lymphocytes were extracted for flow cytometry analysis, as previously described [65]. Lymphocyte and leukocyte populations were defined using the following markers (clone): CD45 (30F11), CD3E (17A2), CD4 (GK1.5), CD8 (AF700), B220 (RA3-6B2), CD11b (M1/70), and Ly6G (1A8), all from BioLegend. HTR1A was detected using a goat anti-HTR1A antibody (T-16, Santa Cruz, Dallas, TX, USA)) and anti-goat Cy5 secondary antibody, as well as leptin receptor (rabbit anti-ObR, Santa Cruz) and anti-rabbit APC secondary antibody. Data were collected on a BD FACS Canto II using FACSDiva (V6.1.3) software and analysed in FlowJo (V9, Becton Dickinson, Ashland, OR, USA).

In experiments to determine the effect of *H. pylori* infection on the expression of 5-HT receptors, mice were infected by oral gavage, as described above, and were euthanised on day 3 or day 21. Stomachs were collected for RNA extraction and semi-quantitative qPCR to detect the expression of 5-HT receptors (HTR1A, HTR1B, HTR2B). Primer sequences are shown in Appendix A. Gene expression relative to the control, uninfected mice, was determined using a SYBR green assay, and data were normalised to either beta actin or HPRT as the reference genes for human and mouse genes, respectively. Fold changes relative to the control were calculated using the method of Paffl [66]. Bacterial colonisation was determined from DNA extracted from stomach homogenates, and *H. pylori* CFU was determined using a Taq-Man qPCR assay, as previously described [65].

### 4.2. AGS Model of H. pylori Infection

Gastric epithelial carcinoma cell line (AGS) cells (ATCC CRL-1739, passage 5–10) were cultured in RPMI containing 10% foetal calf serum (FCS) and penicillin/streptomycin in the presence of 5% CO_2_. Cultures were passaged every 3 days at 70% confluence. A total of 1 × 10^5^ cells were seeded onto sterile coverslips for co-culture with *H. pylori* and fixed with 4% formaldehyde for microscopy.

*H. pylori* strain 26,695 was cultured on lysed blood agar plates containing a DENT supplement (Oxoid). Bacteria were harvested from plates with a sterile loop and resuspended in Brucella broth containing 10% FCS. *H. pylori* were added to AGS cells at a multiplicity of infection (MOI) of 100. Bacteria were co-cultured with cells for 30, 60, or 240 min at 5% CO_2_. 5-HT-treated cells were treated with 5H-T for 5 min before the addition of *H. pylori.* Cells were washed with ice-cold PBS to remove unattached bacteria, followed by fixation with 4% formaldehyde.

Cells were permeabilised with 0.1% Triton-X 100 (Sigma-Aldrich, St. Louis, MO, USA) in PBS and blocked with 10% goat serum before overnight incubation with primary antibodies (mouse anti-NF-κB p65 Santa Cruz Biotechnology (Sc-8008) and rabbit anti-ERK, Cell signalling Technologies (Danvers, MA, USA) (4695T)). Secondary antibody (Sigma-Aldrich, SAB4600044) and goat anti-mouse IgG H&L CF488 (Sigma-Aldrich, SAB4600042) was used for detection, and cells were counterstained with DAPI and Alexa 565 labelled Phalloidin (Life Science technologies, Leawood, KS, USA) before being mounted in Mowiol 488/10% glycerol in TBS containing DABCO (Sigma-Aldrich) as anti-fade. Imaging was carried out on an A1R-Nikon Confocal microscope (Nikon, Beverly, SA, Australia). Image capture and analysis was carried out using NIS-Elements software (AR 4.10.00).

### 4.3. Carbon Fibre Amperometry

Intestinal 5-HT was measured using an electrochemical technique previously validated in multiple species including mice [67]. Stomachs and segments of the ileum and distal colon were visualised under a dissecting microscope, cut along the greater curvature or mesenteric border, and loosely pinned mucosal-side-up in a silicon-lined recording chamber. The chamber was superfused with carbogen (95% O_2_ and 5% CO_2_) bubbled Krebs solution (composition in mmol L-1: NaCl, 117; NaH_2_PO_4_, 1.2; MgSO_4_, 1.2; CaCl_2_, 2.5; KCl, 4.7; NaHCO_3_, 25; and glucose, 11) at 35 °C at a flow rate of ~5 mL/min and equilibrated for 60 min. Carbon fibre electrodes were prepared as described previously, and voltage clamped at +400 mV; 5-HT oxidation was detected as a positive current deflection. Recordings were made using a VA-10 amplifier (NPI Electronics, Tamm, Germany) digitised at 1–5 kHz (Digidata 1440; Axon Instruments, Union City, CA, USA) to a personal computer using PClamp 9.0 (MDS Analytical Technologies, Mississauga, ON, Canada, 0.5 kHz filtering with a 50 Hz notch filter). Electrodes were calibrated with 5 μL of 5–10 μM serotonin hydrochloride (Sigma-Aldrich, Sydney, NSW, Australia). A micromanipulator was used to compress the mucosa with the carbon fibre microelectrode to induce mechanically stimulated 5-HT release (peak) and the decay of 5-HT back to baseline levels (steady state). The oxidation currents were confirmed using a voltage ramp (0 mV to −500 mV, to +1000 mV, and back to −500 mV before returning to zero). A ramp rate of 275 mV/s allowed for the separation of two clear peaks (Figure 2C). The first signal peaked at +330 mV and corresponded to 5-HT, while the second peak was at ~+750 mV and appeared to be melatonin [68].

### 4.4. 16S rRNA Sequencing Analysis

Total DNA was extracted from stomach contents using the QiaAMP FAST stool DNA mini kit (Qiagen, Clayton, VIC, Australia) and eluted in dH_2_O. All samples were PCR-amplified for 16S rRNA gene analysis. The V3-V4 regions of the 16S rRNA genes were amplified with Q5 high-fidelity polymerase (New England Biolabs, Notting Hill, VIC, Australia) using the 338F (5′-ACTCCTACGGGAGGCAGCAG-3′) and 806R (5′-GGACTACHVGGGTWTCTAAT-3′) primer pairs following the dual barcoding method, as previously described [69,70], and sequenced on an Illumina MiSeq instrument using a 2 × 300 bp paired-end kit. Sequencing data analysis was performed using Quantitative Insights Into Microbial Ecology 2 (QIIME2) [71]. Quality filtering, denoising, and chimaera removal were performed using DADA2 [72] as a QIIME2 plugin with all recommended parameters. Taxonomy was assigned using the SILVA v138.1 database [73] for the generation of an amplicon sequencing variant (ASV) table. Downstream statistical microbial data analyses and visualisations were carried out using MicrobiomeAnalyst, where features with mean values of <10 were filtered [74].

A total of 573,977 sequences were obtained from 25 samples, with an average of 22,959 sequences per sample. A total of 4539 ASVs were identified. The sequence data used for analysis are available in NCBI under BioProject accession number PRJNA1221336.

### 4.5. Statistical Analysis

Bacterial colonisation, gene expression, and flow cytometry data were analysed using the Mann–Whitney U test for non-parametric data using GraphPad Prism (version 10.0.0 for Mac, GraphPad Software, Boston, MA, USA, www.graphpad.com). A *p* value of <0.05 was considered significant.

## Figures and Tables

**Figure 1 ijms-26-07735-f001:**
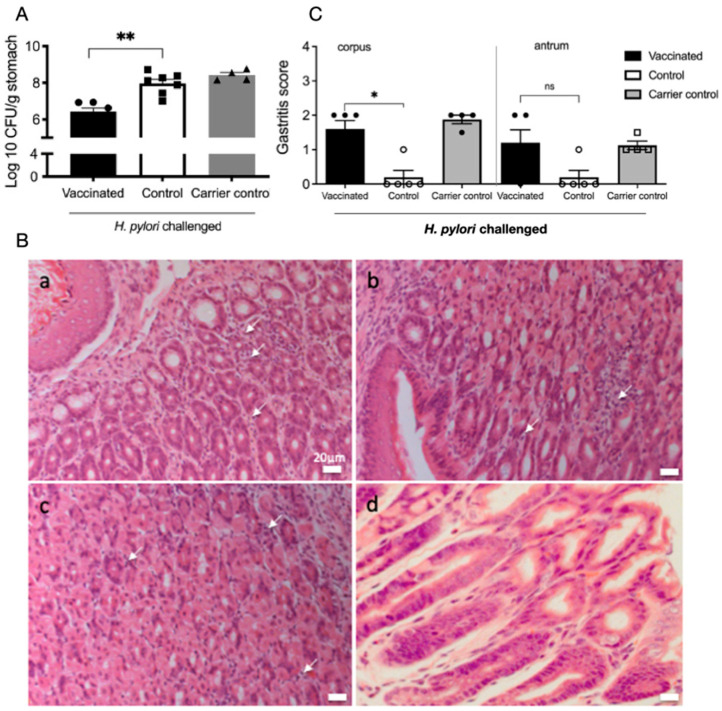
Vaccination leads to reduced *H. pylori* colonisation and increased inflammatory infiltrates in the gastric mucosa. (**A**) Mean CFU in stomachs of mice 21 days after *H. pylori* challenge. Vaccination with a single oral dose of a recombinant-attenuated *S.* Typhimurium expressing *H. pylori* urease leads to significant reductions in colonisation (black bar) compared to sham vaccinated controls (white bars). The salmonella carrier alone (grey bars) did not induce significant reductions *n* = 5 mice/group). (**C**) Inflammation scores are increased in the vaccinated mice compared to controls. Mean gastritis scores in the corpus and antrum. Gastritis scores were significantly higher in the corpus of vaccinated mice (* *p* < 0.05, ** *p* < 0.001, Mann–Whitney U test). Bars represent the mean and SEM, with 3 mice/group. ns not significant (**B**) Stomachs were embedded in paraffin and stained with H&E for histological grading using the modified Sydney system. Representative sections of the upper body at the junction with forestomach from mice at day 21 post-infection; (**a**) vaccinated (gastritis score 3); (**b**) infected (gastritis score 2); (**c**) carrier control (gastritis score 2.5); (**d**) naïve (score 0). White arrows indicate lymphocytic infiltrates. Scale bar 20 μm.

**Figure 2 ijms-26-07735-f002:**
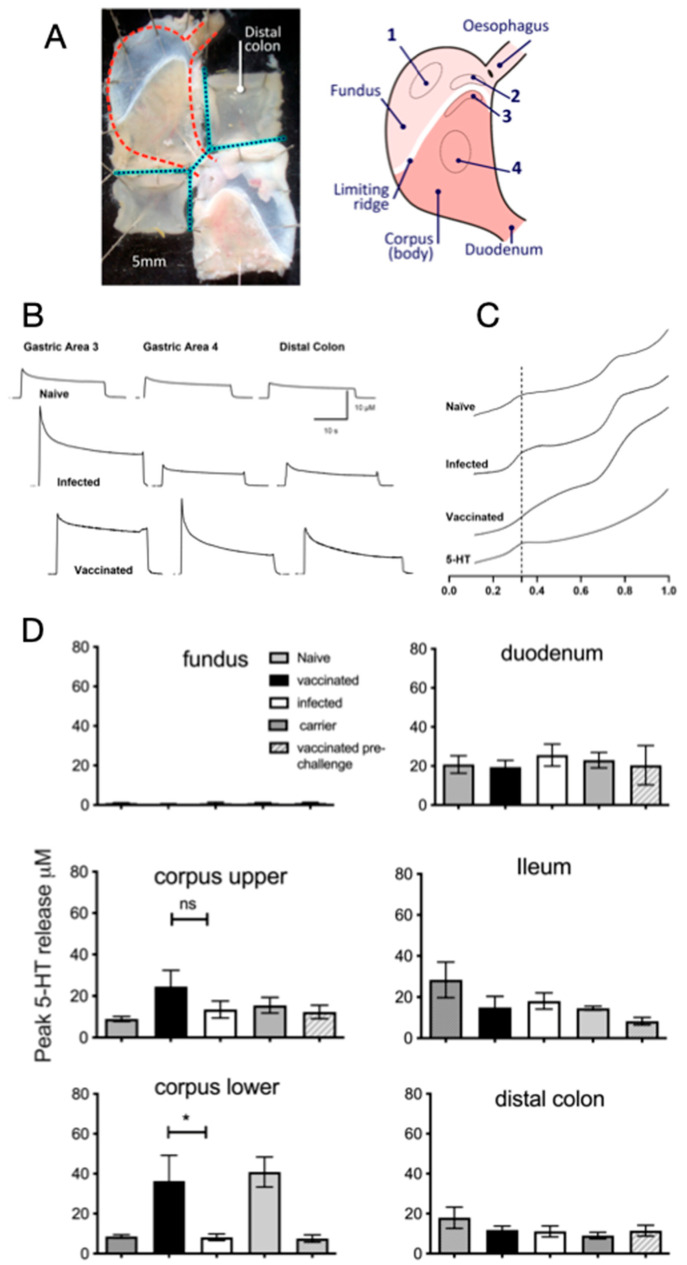
5-HT release as detected by carbon fibre amperometry and increased release in the corpus region of vaccinated, *H. pylori*-challenged mice. (**A**) Left: A top-down photo of tissues from two animals pinned in an organ bath for amperometric recording. The dashed red line indicates the approximate shape of the stomach prior to dissection. Dotted black lines show the separation of stomachs and distal colons. Right: Diagram of a mouse stomach showing the anatomical regions and recording areas (1–4). (**B**) Representative amperometry traces from areas 3, 4, and the distal colon in naïve (N, top), infected (I, middle), and vaccinated (V, bottom) mice. (**C**). Voltage ramp traces from area 3 in naïve, infected, and vaccinated mice. Dotted line is at 330 mV, with the expected oxidation potential of 5-HT; the second peak was likely to be melatonin but was not investigated further. (**D**) Mean peak 5-HT release in the analysed regions of the stomach, duodenum, ileum, and distal colon. Increased 5-HT release was detected in the lower corpus of vaccinated and carrier vaccinated mice. Data representative from two independent experiments. (* *p* < 0.05, ns. not significant, Mann–Whitney U test.) Bars represent the mean and SEM, with five mice/group.

**Figure 3 ijms-26-07735-f003:**
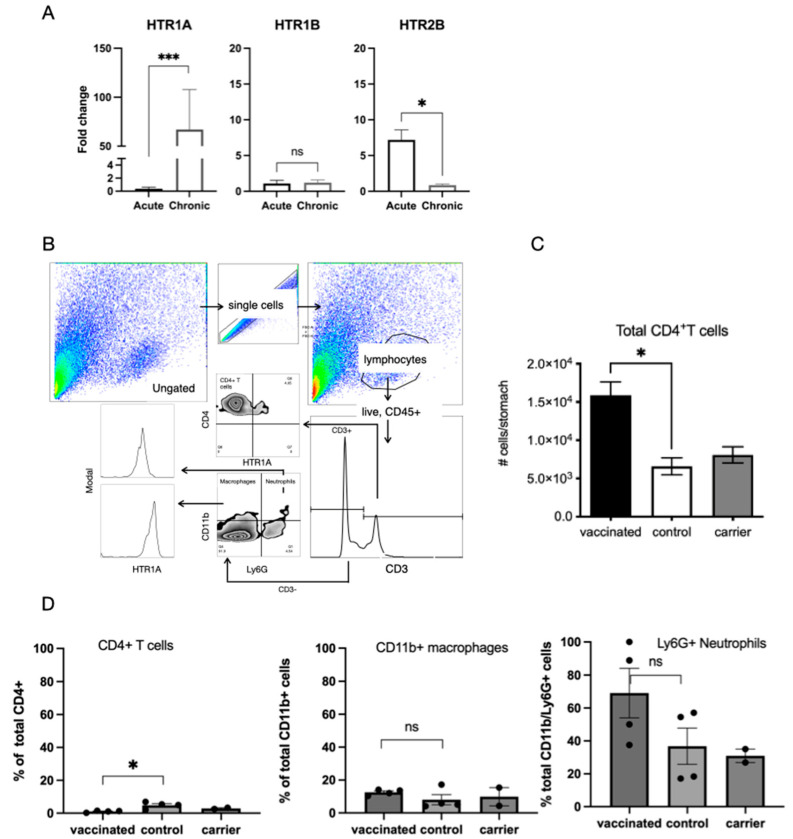
Expression of 5-HT receptors in the gastric mucosa and immune infiltrates in *H. pylori*-challenged mice. (**A**) qPCR analysis of the expression of 5-HT receptors HTR1A, HTR1B, and HTR2B in the gastric mucosa. RNA was extracted from the stomachs of C57BL/6 mice infected with *H. pylori* SS1 for 3 days (acute infection) or 3 weeks (chronic infection). Expression levels were determined by qPCR and are expressed as a fold change relative to naïve mice (*n* = 5 mice/group). HTR1A was increased in chronically infected stomachs, whereas HTR2B was increased in acutely infected stomachs. (* *p* < 0.05, *** *p*<0.0001, ns. not significant). (**B**) Gating strategy for FACS analysis of gastric infiltrates from *H. pylori*-infected mice after 3 weeks of infection reveals the expression of HTR1A on CD4+ T cells, macrophages, and neutrophil populations, with representative plots from five mice/group. (**C**) Increased infiltrates of CD4+ T cells in vaccinated compared to control mice (*p* < 0.05). (**D**) The proportion of positive HTR1A on inflammatory cell populations in mice after 3 weeks. Expression was reduced in vaccinated compared to control mice.

**Figure 4 ijms-26-07735-f004:**
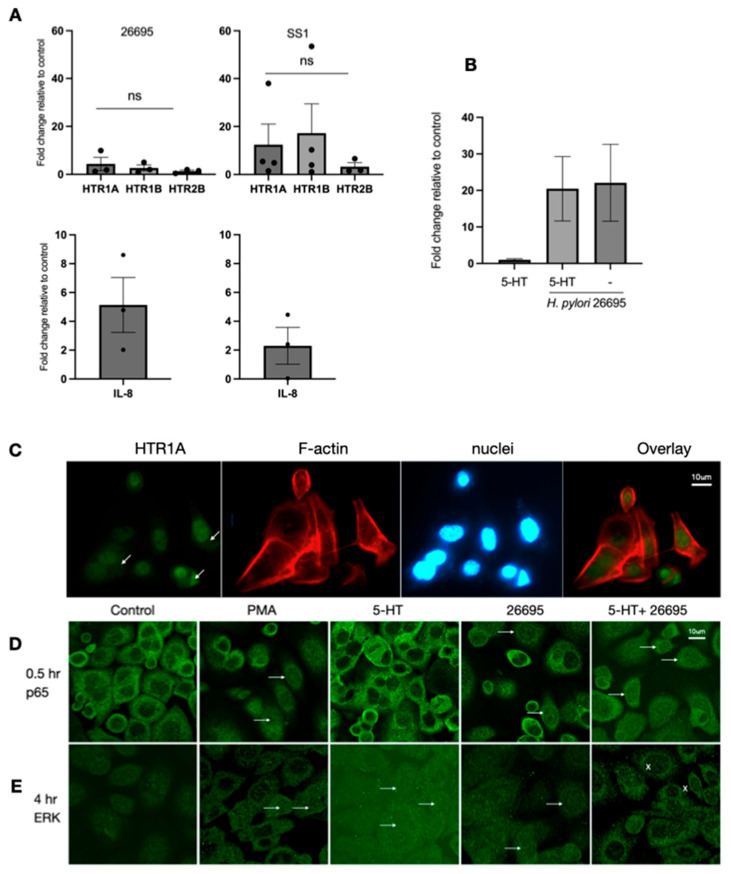
Effect of *H. pylori* infection on gastric epithelial cell responses to 5-HT. (**A**) The AGS gastric epithelial cell line was co-cultured with *H. pylori* 26,695 for 6 h, and the expression of 5-HT receptors (top) and IL-8 (bottom) was determined by qPCR. (**B**) Pretreatment with 5-HT for 5 min did not affect the induction of IL-8 expression by *H. pylori* after 6 h. (**C**) The expression of HTR1A protein in AGS cells exposed to *H. pylori* for 6 h. HTR1A (green) was distributed in the cytoplasm and nuclear regions (arrows), nuclei (blue), and F-actin (red). (**D**) Activation of NF-κB as reflected by the nuclear translocation of the p65 subunit 0.5 h after exposure to *H. pylori*, with or without 5-HT (arrows). Exposure to *H. pylori* induced NF-κB translocation but not after treatment with 5-HT alone. 5-HT treatment did not increase *H. pylori*-induced NF-κB activation. (**E**) Activation of ERK as reflected by nuclear translocation in AGS cells exposed to *H. pylori* for 4 h. Both PMA and *H. pylori* exposure caused nuclear translocation (white arrows), but translocation was blocked by combined 5-HT and *H. pylori* (x) (representative results from three independent experiments).

**Figure 5 ijms-26-07735-f005:**
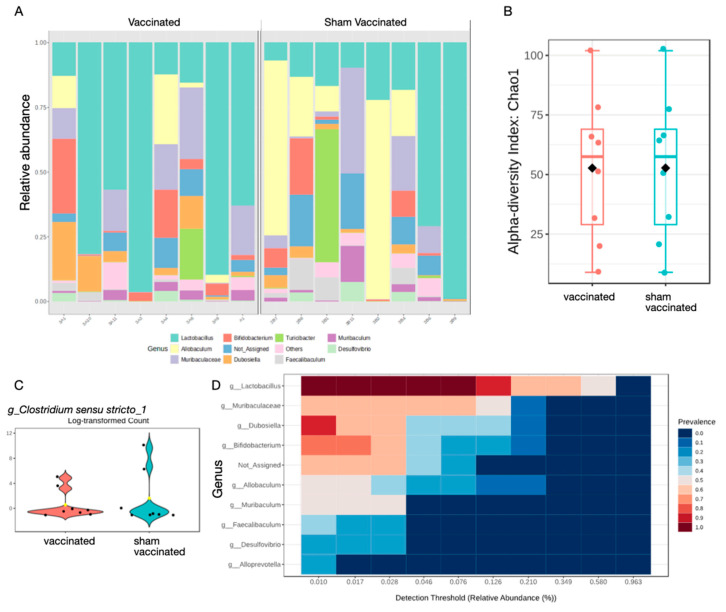
Effect of *H. pylori* infection and vaccination on gastric microbiota. DNA was extracted from the gastric contents of vaccinated and sham vaccinated *H. pylori*-challenged mice at day 21 post-challenge and processed for 16S RNA gene sequencing. (**A**) *H. pylori* infection or vaccination did not cause an overall significant change in relative abundance or (**B**) alpha diversity (genus level, Chao1). (**C**) Abundance of *Clostridium* sensu stricto genus was significantly higher in sham vaccinated mice, and (**D**) the core microbiome of vaccinated mice was biassed toward *Lactobacilli*.

**Figure 6 ijms-26-07735-f006:**
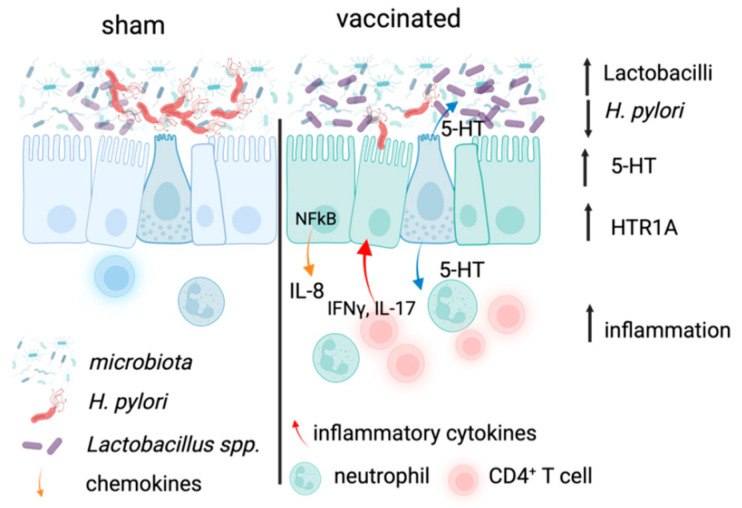
Summary of proposed role for how vaccination drives 5-HT release in the context of *H. pylori* vaccination. In sham vaccinated stomachs, *H. pylori* induces mild inflammation that is not protective. In vaccinated, *H. pylori*-challenge stomachs, specific CD4 T cells rapidly infiltrate the gastric mucosa, inducing protective responses; the inflammatory state increases the secretion of 5-HT and the expression of HTR1A, potentially impacting the function of both epithelial cells and neutrophils. The combined effects reduce *H. pylori* colonisation and promote *Lactobacillus* spp. in the lumen. Created in BioRender. Walduck, A. (2025) https://BioRender.com/rv9rkue (accessed on 4 August 2025).

## Data Availability

The microbiota dataset from this study is accessible at NCBI Bioproject https://www.ncbi.nlm.nih.gov/bioproject/?term=PRJNA1221336 (accessed on 9 February 2025).

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
