# Peer review of "Gastric Inflammation Impacts Serotonin Secretion in a Mouse Model of Helicobacter pylori Vaccination†"

_ijms, 2025, doi:10.3390/ijms26167735_

Round 1
Reviewer 1 Report
Comments and Suggestions for Authors
This study of Idowu et al, represents an interesting and novel exploration of the role of serotonin in gastric inflammation and microbiota modulation during H. pylori infection. While the authors suggested that an increased serotonin release in the gastric corpus of vaccinated mice is therefore likely to be a result of increased local inflammatory infiltrates rather than a role for the gastric microbiota a more detailed mechanistic exploration would strengthen the impact of the manuscript.
Minor revisions:
Terms 3w and 3d (weeks and days) should be clearly defined within the abstract.
Please improve the writing of section 2.4 (lines 175-187). Besides, Figure 4D is referenced before Figure 4C. No proper reference either for Figure 4C nor Figure 4E. Also, add space in “6hours” (line 176).
Fig 5B line 207 or Figure 5 C in line 216. Please keep consistent.
GC abbreviation (line 217) is missing in Abbreviations’ section.
Typo: should be Panel D instead E in line 226.
Format of Figure 5’s legend is different from the other legends. Please keep consistent.
Diarrhoea (typo in line 295)
More details in Methodology. Missing information about fluorescence microscopy.
All data not shown statements found in the manuscript (i.e, in lines 208-209) are not acceptable. Please provide as Supplementary Information data.
Core microbiome of vaccinated mice was biased toward Lactobacilli. Please discuss about Lactobacilli relevance in this context.
Author Response
Comment 1: Terms 3w and 3d (weeks and days) should be clearly defined within the abstract.
Response 1: Text has been revised.
Comment 2: Please improve the writing of section 2.4 (lines 175-187). Besides, Figure 4D is referenced before Figure 4C. No proper reference either for Figure 4C nor Figure 4E. Also, add space in “6hours” (line 176).
Response 2: The text has been revised. We have now added a reference for Figure 4C (lines 218-219, including the addition of space in “6 hours” (lines 220-227).
Comment 3: Fig 5B line 207 or Figure 5 C in line 216. Please keep consistent. Different methods of evaluating diversities were employed. While the Chao1, Welch T-test/ANOVA and Shannon diversity index revealed no change.
Response 3: For clarification, line 207 was talking about the lack of overall changes in diversity while line 216 was about single factor analysis that revealed the increased abundance of Clostridium senso stricto 1. This was not enough to cause significant changes in overall diversity.
Comment 4: GC abbreviation (line 217) is missing in Abbreviations’ section.
Response 4: This has now been added into the abbreviations section.
Comment 5: Typo: should be Panel D instead E in line 226.
Response 5: The error has been corrected. (Line 266)
Comment 6: Format of Figure 5’s legend is different from the other legends. Please keep consistent.
Response 6: Figure legend has been modified.
Comment 7: Diarrhoea (typo in line 295).
Response 7: Diarrhoea in line 302 of the submitted manuscript (now in line 359) is spelt correct (Australian English spelling).
Comment 8: More details in Methodology. Missing information about fluorescence microscopy.
Response 8: Details about fluorescence microscopy was written as part of section 4.2 (line 347-353) of the submitted manuscript. Information about the microscope used had been omitted in error and now added in lines 419-421.
Comments 9: All data not shown statements found in the manuscript (i.e, in lines 208-209) are not acceptable. Please provide as Supplementary Information data.
Response 9. The data for these items were not shown because the lack of significant difference did not add to the interpretation of the data. Supplementary Figure S1 now contain this data for reference.
Comments 10: Core microbiome of vaccinated mice was biased toward Lactobacilli. Please discuss about Lactobacilli relevance in this context.
Response 10: The text has been modified. We have now emphasized the relevance of Lactobaccilli in increasing H. pylori eradication rate (318-320). Additional remarks have also been added in response to a comment by Reviewer 2.
Reviewer 2 Report
Comments and Suggestions for Authors
This study investigated the role of serotonin (5-HT) in the gastric response to H. pylori infection and prophylactic vaccination in mice. A significant finding was that vaccination, which reduced H. pylori colonization, led to increased local 5-HT release specifically in the gastric corpus, correlating with elevated inflammatory infiltrates. Immune cells like CD4+ T cells, macrophages, and neutrophils within gastric infiltrates were shown to express HTR1A, although vaccinated mice had a lower proportion of CD4+ T cells expressing this receptor. In vitro experiments using gastric epithelial cells (AGS) revealed that while 5-HT did not enhance H. pylori-induced NF-κB activation, it effectively blocked ERK translocation, suggesting a potential role in maintaining homeostasis. The study also examined the gastric microbiota, noting that vaccination did not significantly alter overall diversity but showed a bias towards Lactobacilli and decreased Clostridium sensu stricto 1 in vaccinated mice compared to controls. The observed increase in gastric 5-HT appears to be a consequence of the local inflammatory response rather than microbiota changes at the tested timepoint.
This is more than interesting and systematic research that sheds light on the role of serotonin in the gastric immune response to H. pylori and the effects of vaccination, with perspectives for new treatment strategies.
Some remarks and some recommendations can be added in the comments in the attached annotated paper manuscript.

Author Response
Comments on attached document
Comment 1: The title should be reformulated so that, in one way or another, it suggests the role of serotonin in the inflammatory response. "Investigating" is not a term that impacts the reader.
Response 1: The title has been modified.
Comment 2: Abstract. This paper is a research article. The abstract should be organized in an "usual" biomedical research paper abstract, i.e.: Background / Context, Objectives, Materials and methods, Results and Conclusions. The text is already in the abstract; the authors simply need to insert the above separators.
Response 2: Subheadings in the abstract are not part of IJMS format so we will defer to the editor on this point. In the revised version the abstract (Line 14-30) has been modified to include the separators- Background, Methods and Objectives, Results, and Conclusion.
Comment 3: Line 32. Introduction. Though this phrase is true, it is difficult to identify the link with the previous introductory phrase. Could you please rephrase and/or explain?
Response 3: The text has been modified We have now included additional remarks in lines 36-41 (highlighted) to explain what we meant.
Comment 4: Introduction. Line 37. The term "cytokine" usually implies immune cell-derived, paracrine-acting proteins. Gut hormones are endocrine peptides, and while they may have cytokine-like effects, they are not cytokines by definition. So, I would recommend rephrasing to imply "cytokine-like behaviour"
Response 4: The text has been modified. We have now replaced it with the phrase “cytokine-like behaviour” in the manuscript (Line 44).
Comment 5: Line 48. This phrase is not complete.
Response 5: The text has been modified. We have now added an additional phrase (highlighted) at the end of the sentence (Line 55).
Comment 6: Line 51, 58-60. And as far it can be understood from your paper you are addressing this issue - you are investigating the serotonin role in mediating the immune response.
Response 6: Yes. The text has been modified.
Comment 7: Line 56. Shouldn't be 5-HT1RA?
Response 7: The correct notation is 5-HT1A receptor.
Comment 8: Lines 67-69. This sounds more like a phrase for the "Discussion" or for the "Conclusions" section.
Response 8: We have not modified the text because we used that to build our argument as a broad reason why this study was carried out. More specific objectives and actual experiments carried out in the study had been mentioned in the discussion (lines 233-239, now lines 273-279).
Comment 9: Lines 75-78. This should be slightly corrected.
Response 9: The text has been modified. An additional phrase has now been added (highlighted) to emphasise the point being made (Lines 84-85).
Comment 10: Lines 78-79. Whose low sensitivity?
Response 10: The text has been modified to emphasize that the sensitivity of the methods is what is being referred to (Line 86).
Comment 11: Lines 88-94. Your are building nicely the knowledge gaps in literature which you are addressing in this study. I think this phrase should be more articulated in terms of what gaps your are trying to approach and how (the research and the paper organisation).
Response 11: The text has been modified. We have now expanded the first sentence of the paragraph to further specify the knowledge gaps being filled (Line 107-109).
Comments 12: Some other points that should be added (Section 2.1:
- The corpus shows a significant increase in inflammation in vaccinated mice; the antrum does not (Fig. 1C). Response: The text has been modified. We have now added a statement in that section (highlighted) that highlights this point (Line 126). The figure legend has also been modified (highlighted) to emphasize this point (Line 137).
- Inflammatory cell infiltrates are visibly denser in vaccinated mice (panels a–c), while panel d (naive) shows intact mucosa. (Fig 1B). Response: The text has been modified. We have now added a statement in that section (highlighted) that highlights this point (Lines 124-125).
- The carrier control group shows no protection (Fig 1.A) and low inflammation (Fig 1. C). Response: The text has been modified We have now added a statement in that section (highlighted) to emphasize this point (Lines 120-121).
Comments 13: Some suggestions for section 2.2
1) Only the lower corpus shows vaccine-induced 5-HT elevation with no significant changes observed in the upper corpus, suggesting a more precise regional specificity of vaccine-induced activation. Response: Thanks for the constructive remarks. The text has been modified. We have now added a few more phrases in the section to highlight this point (Lines 150-152).
2) The absence of increased 5-HT release in the carrier control group highlights that the observed effect is antigen-specific and not attributable to the carrier strain alone. Response: The text has been modified. We have now added a few more phrases in the section to highlight this point (Lines 153-155).
3) Although 5-HT levels in the duodenum were modestly elevated in the vaccinated group, this trend was not statistically significant, further reinforcing that the vaccine-induced effect is primarily localized to the gastric corpus. Response: We disagree as there was no such elevation in the duodenum of the vaccinated group.
4) Electrochemical traces (Figure 2B) support these findings, with visibly enhanced 5-HT peaks in the lower corpus of vaccinated animals, aligning with quantitative measurements. Response: The text has been modified. We have now added a few more phrases in the section to highlight this point (Lines 157-159).
Comments 14: Some suggestions for additions / modifications (Section 2.3)
1) Among immune cell populations, CD4⁺ T cells exhibited the highest proportion of HTR1A expression, suggesting a potentially more prominent serotonergic signaling role in T cell-mediated immune modulation compared to myeloid cells. Response: 5-HT has been shown to signal and affect function of both myeloid and T cell lineages (Arreola et al., 2015), so the observations are likely limited to the context of vaccination. The text has been modified. We have now added an additional statement to emphasize this (185-189).
2) The selective expansion of CD4⁺ T cells in vaccinated, but not carrier-treated, mice confirms that the response is antigen-specific and not an adjuvant or vector-related effect. Response: The text has been modified. We have now added phrases in the section to emphasize this (190-192).
3) The gating strategy (Figure 3B) effectively distinguishes lymphoid and myeloid populations, allowing robust identification of HTR1A expression across distinct immune subsets. Response: The text has been modified. We have now added an additional statement to emphasize this (184-185).
4) The high inter-individual variability observed in HTR1A expression among neutrophils suggests that regulation may be context-dependent or subject to local microenvironmental cues. Response: Thank you for the constructive comment. We have now added an additional statement to emphasize this (198-200).
Comment 15: Neutrophils show the trend, but macrophages do not.
Response 15: We agree, however the differences were not statistically different. The text has been modified. We have now added an additional comment to the sentence to emphasize this (197-198).
Comment 16: No reference is made to Figure 4C.
Response 16: The text has been modified. Response has now been made to Figure 4C (Line 218-220).
Comment 17: Although 5-HT alone did not activate NF-κB, its presence during H. pylori exposure appeared to attenuate p65 nuclear translocation, suggesting a partial inhibitory effect on NF-κB activation. This helps nuance the phrase "did not increase," which currently overlooks a potentially relevant suppressive role.
Response 17: We did not detect any evidence of a potential suppressive role of NFkB activation. We however, added a hyphen (highlighted) in line 223 to further emphasize and clarify our points to the reader.
Comment 18: IL-8 induction shows high variability but no additive or synergistic effect when 5-HT is added to H. pylori??? Missing comparative test.
Response 18: The mRNA levels of IL-8 were more variable than expected, however this may reflect the fact that 5-HT pre-exposure was 6 hour prior. IL-8 expression is downstream of NF-κB activation and the lack of additive effect of 5-HT shown in panel B is however consistent with the observed lack of additive effect of 5-HT on NF-κB activation shown in panel D. The observed effect on ERK translocation confirms that the 5-HT exposure used was sufficient to trigger signalling events.
Comment 19: HTR1A signal (green) localizes to both cytoplasmic and possibly nuclear/perinuclear regions in AGS cells.
Response 19: Yes. The text has been modified. This was only mentioned in the figure legend initially, but we have now added an additional phrase to emphasize this point in the main section (Lines 219-220).
Comment 20: Nuclear translocation of p65 is visibly reduced in the 5-HT + H. pylori condition compared to H. pylori.Not absent, but less intense.
Response 20: We saw no evidence of reduced translocation in the 5-HT+ H. pylori condition. It would be overinterpretation to conclude any reduction from our data. The nuclear translocation was at the same intensity in some of the other cells in the image, although we might not have pointed the arrows at them.
Comment 21: So, this genus is significantly elevated in sham mice. How this connects microbiome findings to potential host health implications, i.e. gastric inflammation and cancer risks (eventually to be added in Discussions).
Response 21: The text has been modified. We have now further emphasized on its relationship with gastric cancer and inflammation (Lines 329-330).
Comment 22: Is there any stabilizing effect of the vaccination? Sham vaccinated mice displayed higher inter-individual variability in microbial composition, suggesting a more unstable or less uniform gastric microbiota compared to vaccinated mice, in which Lactobacillus was consistently dominant.
Response 22: Thanks for the constructive remark. The text has been modified. (Lines 318-320).
References
Arreola, R. et al. (2015) ‘Immunomodulatory Effects Mediated by Serotonin’, Journal of Immunology Research, 2015(1), p. 354957. Available at: https://doi.org/https://doi.org/10.1155/2015/354957.
Reviewer 3 Report
Comments and Suggestions for Authors
In the article “Investigating the role of serotonin in inflammatory responses induced by H. pylori infection”, the authors investigate the expression of serotonin receptors and serotonin levels in stomachs and tissue culture systems in response to H. pylori infection. While the data represented by this article is intriguing, the emphasis the authors are attempting to put on the difference in immune response after vaccination versus that occurring under normal infection needs clarity. There are other changes that I recommend prior to acceptance of this manuscript. Please see below.
- The title of this article is misleading as there is not a focus on the inflammatory response to H. pylori infection. Instead, it seems to this reviewer that instead the focus is on how vaccination is changing the immune response as compared to what occurs under normal infection and that investigation of the nature of these changes can inform on potential therapeutics/preventatives. Please consider changing the title to something more fitting.
- Along these lines, there needs to be more information about the type(s) of H. pylori vaccinations that are available along with their advantages and disadvantages. It is not until much further in that it’s even mentioned that vaccination is only successful in the mouse model. Something else that needs to be addressed is that the vaccination strategy used in this paper is attenuated S. typhimurium expressing H. pylori proteins. S. typhimurium is a mouse pathogen, and even with attenuation potentially causes shifts in the immune response that are not seen in the control animals. Please include a couple of sentences addressing why this is or is not a concern. Is this the same vaccination that has been attempted in humans?
- Line 89 states that only in vitro infection models were used, please change to in vivo and in vitro.
- Lines 101-103, this statement doesn’t make sense. The data does not show reduced colonization in the carrier control (Figure 1A).
- Figure 1. Please change C. to B. and B. to C. to help with flow of looking at the image. Also please change lower case letters to number to better distinguish labeling of the images. Include enumeration of how many images were scored to give the data represented in the graph.
- I am concerned that the number of mice used is not enough to truly generate statistically relevant data. Please include a power analysis in the materials and methods to demonstrate that this number is sufficient.
- Section 2.2. Mentions electrochemical measurements done on the duodenal tissue but that data is not shown or mentioned further. Either include the data and signatures and explain results or don’t include it. Would be interesting to know those results whatever they were though.
- Figure 2D. I can understand why the scale on all of the graphs was kept the same, but it does not allow the reader to see if there were any changes trend wise to be seen in the fundus. Please include a table with the actual numbers or change the scale on the charts so that the results can be seen.
- Also Figure 2D, why were only statistics between vaccinated and infected examined? What about between naïve and vaccinated or vaccinated and carrier? Also, in the corpus lower, the carrier is as high as the vaccinated. Would this not suggest that it is the S. typhimurium that could be driving serotonin release?
- Section 2.3. It is confusing as to whether the authors are performing a different experiment here. Are the Acute and Chronic mice also vaccinated? If so, is it possible the vaccine is causing the upregulation of receptors and driving the infiltrates? Please clarify what is being described here. If the acute and chronic samples were not from vaccinated mice, how do the authors now that the HTR expression is the same in mice that are vaccinated?
- Section 2.4 needs much more description. Why were the selected timepoints chosen? Were any other timepoints measured? Why were NFkB and ERK chosen and not any other inflammatory pathways (besides IL-8)?
* as this data comes from a tissue culture model, the authors should perform more experiments and for longer incubation periods. Three measures is not sufficient. Especially as what is marked as ns in 26695 infected cells actually appears to be trending towards something. Please also make clear in the text that this is a measure of transcription. To determine that the protein levels are actually doing something, one would need western blots or immunofluorescence.
* The IL-8 data is odd. In A, expression is increased after H. pylori incubation after 6 hrs., but in B that level is MUCH higher (with and without 5-HT). How do the authors explain this? Why more measures would help and looking at protein with ELISA or other means.
* The text states that pretreatment with 5-HT was for 5 minutes (Figure 4B) whereas the legend says it was for 1 hour, please clarify and describe why the specific preincubation time was used.
* for the immunofluorescence, a graph needs to be included to demonstrate the number of cells analyzed (and number of fields of view) to determine the level of translocation that occurs. Again, are the incubation periods are sufficient?
* Localization of HTR1A in the cytoplasm is concerning. Isn’t this supposed to be an outer membrane protein?
- Line 217, define the acronym GC
- Figure 5. To investigate the difference between the effects of H. pylori vaccination on gastric microbiota, the authors would need to have examined the control mice microbiota (no vaccination/sham vaccination) to ensure that the S. typhimurium sham is not initiating changes on its own.
- Line 232 should also be in the introduction
- Lines 233-235, doesn’t this statement effectively argue that looking at the immune response in the animal model does NOT inform on what is occurring in humans?
- The results with HTR1A don’t make sense to me where it’s shown that this receptor increases in vaccinated stomachs but is decreased on CD4+ T-cells. Where is the increased HTR1A found then?
- Line 265, not exactly true. The data showed that transcription increased but would need other, protein specific results to guarantee that protein expression was up.
- Is the vaccination strategy described in this paper always how vaccination is done or are there other strategies?
- Please include a summary figure of how the authors hypothesize vaccination drives HTR expression and serotonin secretion and response. It will help the reader to better understand the data being presented.
Author Response
Reviewer 3
In the article “Investigating the role of serotonin in inflammatory responses induced by H. pylori infection”, the authors investigate the expression of serotonin receptors and serotonin levels in stomachs and tissue culture systems in response to H. pylori infection. While the data represented by this article is intriguing, the emphasis the authors are attempting to put on the difference in immune response after vaccination versus that occurring under normal infection needs clarity. There are other changes that I recommend prior to acceptance of this manuscript. Please see below.
Comment 1: The title of this article is misleading as there is not a focus on the inflammatory response to H. pylori infection. Instead, it seems to this reviewer that instead the focus is on how vaccination is changing the immune response as compared to what occurs under normal infection and that investigation of the nature of these changes can inform on potential therapeutics/preventatives. Please consider changing the title to something more fitting.
Response 1: The title has been modified.
Comment 2: Along these lines, there needs to be more information about the type(s) of H. pylori vaccinations that are available along with their advantages and disadvantages. It is not until much further in that it’s even mentioned that vaccination is only successful in the mouse model. Something else that needs to be addressed is that the vaccination strategy used in this paper is attenuated S. typhimurium expressing H. pylori proteins. S. typhimurium is a mouse pathogen, and even with attenuation potentially causes shifts in the immune response that are not seen in the control animals. Please include a couple of sentences addressing why this is or is not a concern. Is this the same vaccination that has been attempted in humans?
Response 2: The text has been modified to provide further information on this vaccination model. We have now added an additional paragraph into the introduction where we explained types of H. pylori vaccinations and how the safety of the vaccine employed in this study has been shown in mice and humans, and how its efficacy in mice has been previously demonstrated. We did not write comprehensively on different vaccine strategies in order not to derail from the focus of this paper (Lines 95-106).
Comment 3: Line 89 states that only in vitro infection models were used, please change to in vivo and in vitro.
Response 3: The text has been modified. We have now added in vivo and in vitro (lines 108-109).
Comment 4: Lines 101-103, this statement doesn’t make sense. The data does not show reduced colonization in the carrier control (Figure 1A).
Response 4: The text has been modified for clarification. We have now corrected the statement (Lines 120-121).
Comment 5: Figure 1. Please change C. to B. and B. to C. to help with flow of looking at the image. Also please change lower case letters to number to better distinguish labelling of the images. Include enumeration of how many images were scored to give the data represented in the graph.
Response 5: The labelling of the figure panels has been modified. Scoring: The 10 fields from 40 x magnification from n=3 mice were enumerated to determine the inflammation scores. The increased inflammatory scores in vaccinated mice results are consistent with the results from previous studies by our group (eg. (Kaparakis et al., 2008)) and other authors (Sutton et al., 2001).
Comment 6: I am concerned that the number of mice used is not enough to truly generate statistically relevant data. Please include a power analysis in the materials and methods to demonstrate that this number is sufficient.
Response 6:
The number of animals required per group is well- established when working with the H. pylori vaccination model, we have referred to a number of previous studies with groups sized between 5 and 10/ group. However, a power calculation is provided here below for information. A remark has been added to the text (line 364).
The numbers of animals required in each experimental group are determined based on our experience and from similar studies from the scientific literature. Given that immunisation does not induce sterile immunity in mice, the outcome of these experiments is determined on the basis of a statistically significant reduction (p<0.05) in bacterial numbers in the immunised compared to the infected control group. Bacterial colonization levels are typically reduced 10 to 100 times that of the controls (1-2 log CFU/g) (Bumann et al., 2001; Lucas et al., 2001; Toni et al., 2001; Walduck et al., 2004). Past experience with the mouse model of Helicobacter infection tells us that that colonization levels typically vary between individual animals (average standard deviation 0.4 log CFU*). In order to obtain the statistical power to obtain a significant result, 4-6 mice are required per group. (power calculator: http://www.lasec.cuhk.edu.hk/sample-size-calculation.html )(* values determined from averaging results from 5 independent experiments). We generally use 5 mice per group to determine differences in CFU.
Average CFU immunized*: 6 log CFU/g stomach
Average CFU infected*: 7 log CFU/g stomach
Average SD (Sigma) 0.4
2 sided test- that is test to show that the number of CFU in control group is significantly greater than in immunized group
Power 0.95- 4 per group
Power 0.99- 6 per group
Comment 7: Section 2.2. Mentions electrochemical measurements done on the duodenal tissue but that data is not shown or mentioned further. Either include the data and signatures and explain results or don’t include it. Would be interesting to know those results whatever they were though.
Response 7: The data for duodenal tissue is included in Figure 2D. There was no change under any conditions, so it has not been discussed further. The figure legend has been modified to specifically mention the other regions, as has the text (line 146).
Comment 8: Figure 2D. I can understand why the scale on all of the graphs was kept the same, but it does not allow the reader to see if there were any changes trend wise to be seen in the fundus. Please include a table with the actual numbers or change the scale on the charts so that the results can be seen.
Response 8: The fundus had very low levels of 5-HT under all conditions, thus, comparisons were not possible. The values for 5-HT release were in general >=1.5 uM thus the graph accurately represents the result. The axis has been modified in the Fundus panel to make this clear.
Comment 9: Also Figure 2D, why were only statistics between vaccinated and infected examined? What about between naïve and vaccinated or vaccinated and carrier? Also, in the corpus lower, the carrier is as high as the vaccinated. Would this not suggest that it is the S. typhimurium that could be driving serotonin release?
Response 9: The error of the missing significance bar has been corrected. The amount of 5-HT released in the carrier group was indeed significantly higher compared to control infected mice. This relates to the increased inflammation detected in the carrier group (Figure Fig 1 B). The carrier effect has been previously reported in both mice and humans (Aebischer et al., 2008; Becher et al., 2010) and relates to immune response against the Salmonella carrier. S. typhimurium does not infect the stomach, and the vaccine strain is no longer shed by mice after 30 days post vaccination. There are however likely conserved antigens between the 2 gram- negative pathogens. For this reason, we always include this control group in our studies.
In the context of this study, the increased level of 5-HT in this group in the same gastric region underlines the link between local inflammation and 5-HT release. Additional remarks have been added to the figure legend, Results (line 150) and discussion section ( line 290-293).
Comment 10: Section 2.3. It is confusing as to whether the authors are performing a different experiment here. Are the Acute and Chronic mice also vaccinated? If so, is it possible the vaccine is causing the upregulation of receptors and driving the infiltrates? Please clarify what is being described here. If the acute and chronic samples were not from vaccinated mice, how do the authors now that the HTR expression is the same in mice that are vaccinated?
Response 10: Figures 3A were from separate experiments with unvaccinated mice infected with H. pylori. The results illustrate the effect of the development of gastritis over the course of H. pylori infection, which occurs irrespective of vaccination. The text has been modified. Additional description has been added to the materials and methods (382-385). Figure 3 C is from vaccinated mice at day 21. This time is chosen because previous studies have shown us that the effect of vaccination does not increase after day 21. More phrases have now been added in lines 177 And 180-181 to provide more clarity.
Comment 11: Section 2.4 needs much more description. Why were the selected timepoints chosen? Were any other timepoints measured? Why were NFkB and ERK chosen and not any other inflammatory pathways (besides IL-8)?
Response 11: The text has been modified. A justification for NF-kB has already been provided in lines 45-48 of the submitted manuscript now lines 52-55 in the revised manuscript. Additionally, extracellular signal-regulated kinase (ERK) is upstream of NF-κB and is also independently activated by H. pylori (Seo et al., 2004). IL-8 is the single most upregulated gene in AGS cells infected with H. pylori and is the standard read out in this model. IL-8 has also been shown to have the peak mRNA expression after 6 hours and does not increase further than that (Eftang et al., 2012). Other studies have also reported upregulation of IL-8 mRNA at 6-hour time point (Keates et al., 1997; Bartels et al., 2007).
Additional time points were performed for both NFkB and ERK (0.5-4 hr) as described in the materials and methods (line 410), the timepoints shown were those where an effect was detected.
Comment 12: As this data comes from a tissue culture model, the authors should perform more experiments and for longer incubation periods. Three measures is not sufficient. Especially as what is marked as ns in 26695 infected cells actually appears to be trending towards something. Please also make clear in the text that this is a measure of transcription. To determine that the protein levels are actually doing something, one would need western blots or immunofluorescence.
Response 12: We have now added the word “mRNA” IL-8 and expression in line 218. Also, as stated above; IL-8 is the single most upregulated gene in AGS cells infected with H. pylori and has also been shown to have the peak mRNA expression after 6 hours and does not increase further than that (Eftang et al., 2012). Several other studies have also reported upregulation of IL-8 mRNA after 6-hour time point.
Comment 13: The IL-8 data is odd. In A, expression is increased after H. pylori incubation after 6 hrs., but in B that level is MUCH higher (with and without 5-HT). How do the authors explain this? Why more measures would help and looking at protein with ELISA or other means.
Response 13: The data in panel B are from a separate series of experiments, the overall result of increased expression of IL-8 after exposure to H. pylori is the same as panel A, and consistent with studies as mentioned above. The absolute fold- changes are in a higher range, but still overlapping for the 26695 strain. The lower fold change for IL-8 induction for the H. pylori SS1 compared to 26695 strain is expected as it relates to the fact that the SS1 strain does not translocate the CagA toxin (Eaton et al., 2001; Philpott et al., 2002), which is the chief inducer of NFkB and therefore IL-8 induction in this model. Overall, the induction of IL-8 is completely consistent with other studies, our result that the expression level is not affected by 5-HT treatment is of note and sits with the result for NFkB activation.
Comment 14: The text states that pretreatment with 5-HT was for 5 minutes (Figure 4B) whereas the legend says it was for 1 hour, please clarify and describe why the specific preincubation time was used.
Response 14: The figure legend has been corrected (Line 231), cells were stimulated to 5-HT for 5 minutes (because 5-HT has only short stability) and infected with H. pylori for 6 hours (Line 232).
Comment 15: For the immunofluorescence, a graph needs to be included to demonstrate the number of cells analyzed (and number of fields of view) to determine the level of translocation that occurs. Again, are the incubation periods are sufficient?
Response 15: Yes, the incubation periods are sufficient. Several studies reporting NF-κB activation in H. pylori infection studies have shown its occurrence between 15 minutes and 1 hour, with optimal intensities recorded at 30 minutes to 1 hour in most cases (Noursadeghi et al., 2008; Kang et al., 2023). The number of cells have not been quantified because the effect was clearly reproducible. Results shown are representative of 3 independent experiments.
Comment 16: Localization of HTR1A in the cytoplasm is concerning. Isn’t this supposed to be an outer membrane protein?
Response 16: A previous study has previously reported the cytoplasmic expression of HTR1A (KOPPARAPU et al., 2013).
Comment 17: Line 217, define the acronym GC
Response 17: The text has been modified. GC= gastric cancer. This has now been added to the abbreviation section.
Comment 18: Figure 5. To investigate the difference between the effects of H. pylori vaccination on gastric microbiota, the authors would need to have examined the control mice microbiota (no vaccination/sham vaccination) to ensure that the S. typhimurium sham is not initiating changes on its own.
Response 18: Carrier vaccination does have an effect on the microbiota and this was previously reported by (Aebischer et al., 2006). The results from the inflammation scores, colonisation and FACS analysis show that the vaccine carrier induces non – specific inflammation (that is short lived), but that it does not lead to significant reductions in H. pylori colonisation. This carrier effect was not relevant to the question of a possible relationship between the protective effects of vaccination and the gastric microbiota so has not been included.
Comment 19: Line 232 should also be in the introduction.
Response 19: The text has been modified. This has now been added to lines 95-96 of the introduction.
Comment 20: Lines 233-235, doesn’t this statement effectively argue that looking at the immune response in the animal model does NOT inform on what is occurring in humans?
Response 20: Very few/ no? vaccines developed in a mouse model translate directly to an effective vaccine for humans. Research in the mouse model has however provided considerable insights. The mouse model is a well- accepted model for H. pylori gastritis for over 30 years. The purpose of this study was to investigate serotonin release in a model of vaccination against a gastrointestinal pathogen. The recombinant attenuated Salmonella based H. pylorivaccine was safe an immunogenic in studies in humans (Aebischer et al., 2008) as have a number of other vaccines tested in humans {reviewed in (Walduck and Raghavan, 2019)}. Further the vaccine induced a CD4T cell biased immune response in volunteers, similar to that described in mice. Only 2 vaccine studies have ever been conducted in a human H. pylori challenge model, and they were not significantly protective (Aebischer et al., 2008; Malfertheiner et al., 2008). The overall aim of our work focussing on identifying the mechanisms of vaccine- induced protection is to enable the design of better candidate vaccines.
Comment 21: The results with HTR1A don’t make sense to me where it’s shown that this receptor increases in vaccinated stomachs but is decreased on CD4+ T-cells. Where is the increased HTR1A found then?
Response 21: The expression of HTR1A was increased in the stomachs of mice infected for 21 days (Chronic model) compared to non-infected mice (Fig 3A), these mice are equivalent to the control challenged mice in the vaccination experiment. FACS analysis showed part of this expression was on immune cell populations (Fig 3D). HTR1A is also expressed on epithelial and other cell types and was increased after exposure to H. pylori (Fig 4 A, and C). Based on our results we expect that bulk of the expression was on other cell types in the mucosa that were not detectable in FACS. We tried to perform IHC on cryosections to identify these but the commercially available antibodies were not suitable.
Comment 22: Line 265, not exactly true. The data showed that transcription increased but would need other, protein specific results to guarantee that protein expression was up.
Response 22: The text has been corrected. This has now been clarified as mRNA expression. Line 308.
Comment 23: Is the vaccination strategy described in this paper always how vaccination is done or are there other strategies?
Response 23: The text has been modified. We have now added another paragraph into the introduction section, where this is briefly discussed (Line 95-106).
Comment 24: Please include a summary figure of how the authors hypothesize vaccination drives HTR expression and serotonin secretion and response. It will help the reader to better understand the data being presented.
Response 24: Figure 6 has been added to the manuscript.
References
Aebischer, T. et al. (2006) ‘Vaccination prevents Helicobacter pylori-induced alterations of the gastric flora in mice’, FEMS Immunology & Medical Microbiology, 46(2), pp. 221–229. Available at: https://doi.org/https://doi.org/10.1111/rp10.1016-j.femsim.2004.05.008.
Aebischer, T. et al. (2008) ‘Correlation of T cell response and bacterial clearance in human volunteers challenged with Helicobacter pylori revealed by randomised controlled vaccination with Ty21a-based Salmonella vaccines’, Gut, 57(8), p. 1065 LP-1072. Available at: https://doi.org/10.1136/gut.2007.145839.
Bartels, M. et al. (2007) ‘Peptide-mediated disruption of NFkappaB/NRF interaction inhibits IL-8 gene activation by IL-1 or Helicobacter pylori.’, Journal of immunology (Baltimore, Md. : 1950), 179(11), pp. 7605–7613. Available at: https://doi.org/10.4049/jimmunol.179.11.7605.
Becher, D. et al. (2010) ‘Local recall responses in the stomach involving reduced regulation and expanded help mediate vaccine-induced protection against Helicobacter pylori in mice.’, European journal of immunology, 40(10), pp. 2778–2790. Available at: https://doi.org/10.1002/eji.200940219.
Bumann, D. et al. (2001) ‘Safety and immunogenicity of live recombinant Salmonella enterica serovar Typhi Ty21a expressing urease A and B from Helicobacter pylori in human volunteers’, Vaccine, 20(5–6), pp. 845–852. Available at: https://doi.org/10.1016/S0264-410X(01)00391-7.
Eaton, K.A. et al. (2001) ‘Role of Helicobacter pylori cag region genes in colonization and gastritis in two animal models.’, Infection and immunity, 69(5), pp. 2902–2908. Available at: https://doi.org/10.1128/IAI.69.5.2902-2908.2001.
Eftang, L.L. et al. (2012) ‘Interleukin-8 is the single most up-regulated gene in whole genome profiling of H. pylori exposed gastric epithelial cells’, BMC Microbiology, 12(1), p. 9. Available at: https://doi.org/10.1186/1471-2180-12-9.
Kang, J.H. et al. (2023) ‘IL-17A promotes Helicobacter pylori-induced gastric carcinogenesis via interactions with IL-17RC’, Gastric Cancer, 26(1), pp. 82–94. Available at: https://doi.org/10.1007/s10120-022-01342-5.
Kaparakis, M. et al. (2008) ‘Macrophages Are Mediators of Gastritis in Acute Helicobacter pylori Infection in C57BL/6 Mice’, INFECTION AND IMMUNITY, 76(5), pp. 2235–2239. Available at: https://doi.org/10.1128/IAI.01481-07.
Keates, S. et al. (1997) ‘Helicobacter pylori infection activates NF-kappa B in gastric epithelial cells’, Gastroenterology, 113(4), pp. 1099–1109. Available at: https://doi.org/10.1053/GAST.1997.V113.PM9322504.
KOPPARAPU, P.K. et al. (2013) ‘Expression and Localization of Serotonin Receptors in Human Breast Cancer’, Anticancer Research, 33(2), pp. 363–370. Available at: http://ar.iiarjournals.org/content/33/2/363.abstract.
Lucas, B. et al. (2001) ‘Adoptive transfer of CD4+ T cells specific for subunit A of Helicobacter pylori urease reduces H. pylori stomach colonization in mice in the absence of interleukin-4 (IL-4)/IL-13 receptor signaling’, Infection and Immunity, 69(3), pp. 1714–1721. Available at: https://doi.org/10.1128/IAI.69.3.1714-1721.2001.
Malfertheiner, P. et al. (2008) ‘Safety and Immunogenicity of an Intramuscular’, Gastroenterology, 135, pp. 787–795. Available at: https://doi.org/10.1053/j.gastro.2008.05.054.
Noursadeghi, M. et al. (2008) ‘Quantitative imaging assay for NF-κB nuclear translocation in primary human macrophages’, Journal of Immunological Methods, 329(1–2), pp. 194–200. Available at: https://doi.org/10.1016/J.JIM.2007.10.015.
Philpott, D.J. et al. (2002) ‘Reduced activation of inflammatory responses in host cells by mouse-adapted Helicobacter pylori isolates’, Cellular Microbiology, 4(5), pp. 285–296. Available at: https://doi.org/10.1046/j.1462-5822.2002.00189.x.
Seo, J.H. et al. (2004) ‘Helicobacter pylori in a Korean isolate activates mitogen-activated protein kinases, AP-1, and NF-jB and induces chemokine expression in gastric epithelial AGS cells’, Laboratory Investigation, 84(1), pp. 49–62. Available at: https://doi.org/10.1038/labinvest.3700010.
Sutton, P. et al. (2001) ‘Post-immunisation gastritis and Helicobacter infection in the mouse: a long term study’, Gut, 49(4), pp. 467–473. Available at: https://doi.org/10.1136/gut.49.4.467.
Toni, A. et al. (2001) ‘Immunity against Helicobacter pylori: Significance of Interleukin-4 Receptor α Chain Status and Gender of Infected Mice’, Infection and Immunity, 69(1), pp. 556–558. Available at: https://doi.org/10.1128/iai.69.1.556-558.2001.
Walduck, A. et al. (2004) ‘Transcription profiling analysis of the mechanisms of vaccine‐induced protection against H. pylori’, The FASEB Journal, 18(15), pp. 1955–1957. Available at: https://doi.org/10.1096/fj.04-2321fje.
Walduck, A. and Raghavan, S. (2019) ‘Immunity and Vaccine Development against Helicobacter pylori’, in Advances in experimental medicine and biology. New York: Springer, New York, NY, pp. 1–19. Available at: https://doi.org/https://doi.org/10.1007/5584_2019_370.